# Improving Visual Relation Detection using Depth Maps

## Abstract

We study the effect of using different object information for visual relation detection with a focus on depth maps. State of the art visual relation detection methods mostly rely on object information extracted from RGB images such as predicted class probabilities, 2D bounding boxes and feature maps. In this paper, we argue that the 3D positions of objects in space can provide additional valuable information about object relations. This information helps not only to detect spatial relations, such as *standing behind*, but also non-spatial relations, such as *holding*. Since 3D information of a scene is not easily accessible, we incorporate a pre-trained RGB-to-Depth model within a visual relation detection framework and release a new dataset *VG-Depth* as an extension to Visual Genome. We note that given the highly imbalanced distribution of relations in the available datasets, typical evaluation metrics for visual relation detection cannot reveal improvements of under-represented relations. To address this problem, we propose using *Macro Recall@K* for visual relation detection. Our experiments confirm that the performance of state-of-the-art visual relation detection approaches can significantly be improved by utilizing depth map information.

Relational learning is an established research area in machine learning. State-of-the-art approaches (Nickel et al., 2016) describe relations as triples of the form *(subject, predicate, object)*, such as *(Man, rides, Bike)*. The triples, which all together form a knowledge graph, are typically extracted from structured data such as the infoboxes in Wikipedia and other sources.

In the last years, the detection of relations from images, i.e., visual relation detection, has also gained a lot of interest. One reason is that understanding relations between entities can play an important role in decision making. For example, detecting whether a man is *on* a bike or *next to* a bike is a crucial challenge in autonomous driving. State-of-the-art works in this area utilize information of objects in the scene such as class labels, bounding boxes and RGB features, to capture pairwise relations using different models. In this paper, we argue that relation detection can additionally benefit from objects' 3D information. This information can help to distinguish between many relations such as *behind*, *in front of* and even improve detection in situations where the objects are nearby such as *covered in*.

Unfortunately, most available datasets, specifically the ones with relational annotations such as Visual Relation Detection (VRD) (Lu et al., 2016) and Visual Genome (VG) (Krishna et al., 2017), lack this 3D information since acquiring them is a cumbersome task requiring specialized hardware. We solve this issue by synthetically generating the corresponding depth maps of images in these datasets. Depth maps provide the objects' distance from the camera and the availability of large corpora of RGB-D pairs, e.g. NYU-Depth-v2 (Nathan Silberman & Fergus, 2012) dataset enables us to learn the mapping between any RGB image to its corresponding depth map using a fully convolutional neural network. We can then apply the trained network on images from VG, converting them into depth maps. We call this dataset extension *VG-Depth*[1].

The extracted features from depth maps, together with other object information extracted from the RGB images, are the basis for relation detection in our framework. We note that the typically employed Recall@K metric, which we call as Micro Recall@K, cannot properly reveal the improvements of under-represented relations in highly imbalanced datasets such as Visual Genome. This might be an issue in applications such as autonomous driving where it is important to ensure that the model is capable of predicting also more important but less represented predicates such as *walking on*

---

[1]This dataset will be made publicly available upon publication.

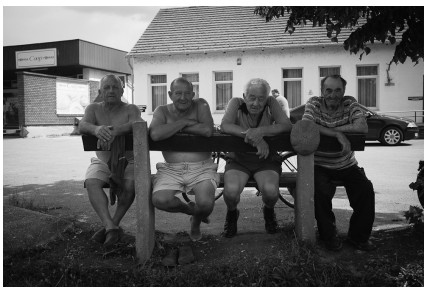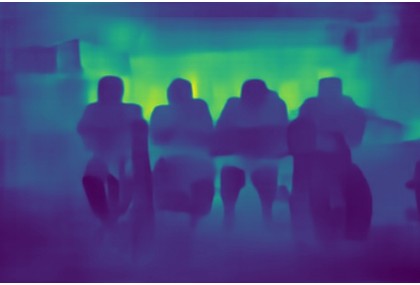

Figure 1: An image from the Visual Genome dataset showing several men sitting next to each other and the corresponding generated depth map. Bright colors indicate a larger distance to the camera. Knowing the depth of, e.g., the individuals, the house and the car, plays an important role in detecting relations such as *next to* and *behind*.

(648 in VG test set) and not just *wearing* (20,148 in VG test set). We tackle this issue by proposing to employ Macro Recall@K, where the Recall@K is computed per predicate class and averaged over, thereby eliminating the effect that over-represented classes have in Micro Recall@K setting. Reporting on both Macro@K and Micro@K, we study the effect of each object information on visual relation detection.

In summary, our contributions are as follows:

1. We perform an extensive study on the effect of different object information in visual relation detection, including 3D information. We show in our empirical evaluations using the VG dataset, that our model can outperform competing methods by a margin of up to $8\%$ points, even those with contextualization or using information extracted from external language sources.
2. We propose using *Macro Recall@K* to better reveal the improvements in visual relation detection when dealing with highly imbalanced datasets such as Visual Genome.
3. To compensate for the lack of 3D data in Visual Genome, we use an RGB-to-Depth model trained on separate corpora of available pairs in both modalities and apply it to VG images and releasing a new dataset, *VG-Depth*.

In general, in this work we aim to answer the following questions:

1. If we are given only depth maps of unknown objects in a scene, how accurately can we infer the distribution of possible pairwise relations? How do other sources of object information compare to it?
2. Current visual relation detection frameworks commonly rely on extensive object information such as class labels, bounding boxes, RGB features, contextual information, etc. Does it bring any additional information, if we also include depth representations in these frameworks or would that only bring redundant knowledge about the scene?
3. Do the current evaluation metrics properly reflect the improvements of under-represented relations within highly imbalanced datasets?

## 1 RELATED WORKS

**Knowledge Graph Learning**  In Knowledge Graph learning, the aim is typically to find embeddings or latent representations for entities and predicates, which then can serve to predict the probability of unseen triples. These methods mostly differ in how they model relations. In RESCAL (Nickel et al., 2011) each relation is defined as a linear transformation in the embedding space of entities, producing a triple probability. TransE (Bordes et al., 2013) employs a similar idea but limits each relation to a translation. In comparison to RESCAL, it has fewer parameters; as a disadvantage, it cannot model symmetric relations. DistMult (Yang et al., 2014) considers each relation as a vector, similar to TransE, but minimizes the trilinear dot product of subject, predicate and object

vector. DistMult can also be understood as a form of RESCAL, where the transformation matrix is diagonal. ComplEx (Trouillon et al., 2016) extends DistMult to complex-valued vectors of embeddings. A multilayer perceptron (MLP) architecture (Dong et al., 2014) extends these methods to non-linear transformations and has shown to be competitive to the other discussed approaches on most benchmarks (Nickel et al., 2016; Socher et al., 2013).

**Visual Relation Detection**    Visual relation detection received a huge boost by the availability of large corpora of annotated images such as the Visual Relation Detection (VRD) (Lu et al., 2016) and the VG (Krishna et al., 2017), containing the visual form of entities, and relations. In VRD, Word2Vec representations of the subject, object, and the predicate were used to train a model jointly with the corresponding image section describing the predicate. In particular, they consider the joint bounding box of subject and object as the image representation for the predicate. Follow-up work achieved improved performance by incorporating a knowledge graph, constructed from the annotated triples in the training set (Baier et al., 2017). In general, the distribution of the predicate bounding box in these works is much more long-tailed than of entities alone. Therefore, separating the models for objects and predicates, as employed in VTransE (Zhang et al., 2017) reduces the complexity of training such a model. VTransE is a generalization of TransE to visual relation detection in which the last convolutional layer of the image detector, together with the location of entities and their class labels, is taken as the input vector to the TransE algorithm. More recently, Yu et al. (Yu et al., 2017) proposed a teacher-student model to distill external language knowledge to improve visual relation detection. Iterative Message Passing (Xu et al., 2017), Neural Motifs (Zellers et al., 2018) and Graph R-CNN (Yang et al., 2018b) incorporate context within each prediction using RNNs and graph convolutions respectively.

**Depth Maps**    While several works have leveraged depth maps to improve object detection (Bo et al., 2013; Eitel et al., 2015; Gupta et al., 2014), the idea of using depth maps in the relation detection task has only been explored in a recent parallel work (Yang et al., 2018a). While the focus of that work is on human-centered relations, we carry out our studies on a broader set of relations (VG), releasing a dataset extension, and also providing a more extensive study on the role of different object information in visual relation detection, with a competent metric.

## 2    FRAMEWORK

In this section we introduce the general framework employed for this study. Let $\mathcal{E} = \{e_1, e_2, ..., e_n\}$ be the set of all entities, including subjects($s$) and objects($o$), and $\mathcal{P} = \{p_1, p_2, ..., p_m\}$ the set of all predicates. Each entity $e_i$ can appear in images within a bounding box $bb_i = (x_i, y_i, w_i, h_i)$, where $(x_i, y_i)$ are the coordinates of the bounding box and $(w_i, h_i)$ are its width and height. In this work we apply the pre-trained (on ImageNet (Russakovsky et al., 2015)) and fine-tuned (on Visual Genome) Faster R-CNN (Ren et al., 2015) on each image $I$ to extract a feature map $\mathbf{fmap_I}$, together with object proposals as a set of bounding boxes $bb$ and class probability distributions $\mathbf{c}$. For each RGB image, we generate a depth map $\mathbf{D}$ where the same bounding box areas encompass the entities' distance from the camera. In the next section, we first describe the employed network for synthetic generation of $\mathbf{D}$s and then discuss the extraction of depth features. In the end, we describe the relation detection head where the late fusion of pairwise features and their relational modelling takes place.

### 2.1    DEPTH MAPS FOR RELATION DETECTION

#### 2.1.1    GENERATION

We incorporate an RGB-to-Depth model within our visual relation detection framework. As shown in Figure 2, this is a fully convolutional neural network (CNN) that takes an RGB image as input and generates its predicted depth map. This model can be pre-trained on any datasets containing pairs of RGB and depth maps regardless of having the object or predicate annotations. This enables us to work with currently available visual relation detection datasets without requiring to collect additional data, and also mitigates the need for specialized hardware in real-world applications. The architectural details are explained in Section 3.

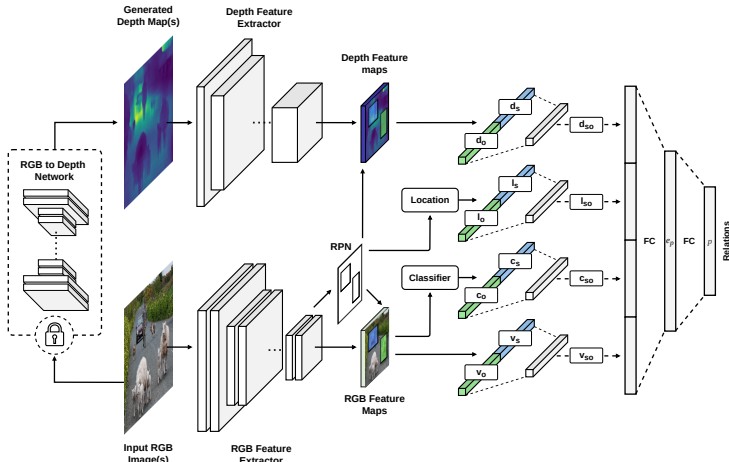

Figure 2: We propose utilizing 3D information in visual relation detection by synthetically generating depth maps using an RGB-to-Depth model incorporated within relation detection frameworks. On the left side, we see the RGB image and its depth map. We use separate CNNs to extract feature maps from both modalities and create pairwise feature vectors $\mathbf{d_{so}}$ (pooled from depth feature maps), $\mathbf{l_{so}}$ (from bounding boxes), $\mathbf{c'_{so}}$ (from class labels) and $\mathbf{v_{so}}$ (from the pooled visual features). These vectors are then concatenated and fed into a relation detection layer to infer the predicate.

### 2.1.2 FEATURE EXTRACTION

Depth maps have been employed in tasks such as *object detection* and *segmentation* (Eitel et al., 2015; Hazirbas et al., 2016). In those works, it is common to simply render a depth map as an RGB image and extract depth features using a CNN pre-trained for RGB images. There, it has been argued that the edges in depth maps might yield better object contours than the edges in cluttered RGB images and that one may combine edges from both RGB and depth to obtain more information (Hazirbas et al., 2016). Therefore, they aimed to get similar, complementary features from both modalities. However, the practice of employing a model pre-trained on a particular source modality, e.g. RGB, and applying it on a different target modality, e.g. depth map, is sub-optimal in many applications[2]. Therefore, similar to other works on depth-based image segmentation or classification we employ a CNN, to generate a feature map $\mathbf{fmap_D}$ from an input depth map, but unlike those works, we train this network from scratch, using depth maps and specifically for relation detection.

### 2.2 RELATION MODEL

In the previous section, we described methods for the extraction of individual object features. Here, we outline the model that infers relations using pairwise combination of these features. For each pair of detected objects within an image, we create a scale-invariant location feature $\mathbf{l_s} = (t_x, t_y, t_w, t_h)$ with: $t_x = (x_s - x_o)/w_o, t_y = (y_s - y_o)/h_o, t_w = \log(w_s/w_o), t_h = \log(h_s/h_o)$ and similarly $\mathbf{l_o}$. We then pool the corresponding features $\mathbf{v_s}$ and $\mathbf{v_o}$ from $\mathbf{fmap_I}$ and create a visual feature vector $[\mathbf{v_s}; \mathbf{v_o}]$. Similarly, we create a depth feature vector $[\mathbf{d_s}; \mathbf{d_o}]$, by extracting $bb_s$ and $bb_o$ from $\mathbf{fmap_D}$ and a class vector $[\mathbf{c_s}; \mathbf{c_o}]$ and similarly $[\mathbf{l_s}; \mathbf{l_o}]$. Each of these vectors are fed into separate layers, yielding $\mathbf{v_{so}}, \mathbf{l_{so}}, \mathbf{c_{so}}$ and $\mathbf{d_{so}}$ before being concatenated altogether and fed to the relation head which projects them to the relation space such that:

$$\mathbf{e_p} = f(\boldsymbol{W}[\mathbf{v_{so}}; \mathbf{l_{so}}; \mathbf{c_{so}}; \mathbf{d_{so}}]). \tag{1}$$

Here, $\boldsymbol{W}$ describes a linear transformation and $f$ is a non-linear function. We realize this as fully connected layers in a neural network with ReLU activations and dropout. $\mathbf{e_p}$ is the embedding vector for predicate which will be learned jointly with other parameters. This simple relation prediction model is a generalization of the model used by (Dong et al., 2014) to construct knowledge graphs.

---

[2]One should also keep in mind that even fine-tuning some layers of a network does not change the very early convolutional filters.

In that paper, latent features for the predicates are also part of the network input, whereas here we use a separate output for each predicate which is easier to train regarding negative sampling, and also has fewer parameters. As shown in earlier works, using more sophisticated models for context propagation between objects with RNNs or graph convolutions, can further improve the prediction accuracy. However, the aim here is to study the effect of including depth maps as additional object representations in visual relation detection and as will be shown later, even with this simple model, they can be a more effective supplement than e.g. propagated context[3]. Clearly, those other models can also further enrich their understanding of object relations, by employing these additional representations.

To learn the parameters, we consider each relation $(s, p, o)$ with an associated Bernoulli variable that takes $1$ if the triple is observed and $0$ otherwise, following a locally closed world assumption (Nickel et al., 2016). Given the set of observed triples $T$, the loss function is the categorical cross entropy between the one-hot targets and the distribution obtained by softmax over the network's output defined as:

$$\mathcal{L} = \sum_{(s,p,o) \in T} - \log \frac{\exp\left(\mathbf{w'}_p^{\mathrm{T}} f(\boldsymbol{W}[\mathbf{v_{so}}; \mathbf{l_{so}}; \mathbf{c_{so}}; \mathbf{d_{so}}])\right)}{\sum_{p' \in \mathcal{P}} \exp\left(\mathbf{w'}_{p'}^{\mathrm{T}} f(\boldsymbol{W}[\mathbf{v_{so}}; \mathbf{l_{so}}; \mathbf{c_{so}}; \mathbf{d_{so}}])\right)} \tag{2}$$

where $\mathbf{w'}_p$ is the weight vector corresponding to $p$ in the last layer (linear classification layer).

## 3 EVALUATION

### 3.1 DATASETS

We test our approach on the *Visual Genome* (Krishna et al., 2017) dataset. We use the more commonly used subset of VG dataset proposed by (Xu et al., 2017) which contains 150 object classes and 50 relations with 75,651 images used for training and (5000 for validation) and 32,422 images for testing.

### 3.2 ARCHITECTURES

**RGB-to-Depth Network:** We use the RGB-to-Depth model architecture introduced in (Laina et al., 2016) which is a fully convolutional neural network built on ResNet-50 (He et al., 2015), and trained in an end-to-end fashion on data from NYU Depth Dataset v2 (Nathan Silberman & Fergus, 2012). Training on the outdoor images from Make3D dataset (Saxena et al., 2007) instead, did not show promising results in our framework. This observation is not surprising since Make3D images contain mostly outdoor scenes with too few objects.

**RGB Feature Extraction:** To extract embeddings and class probabilities of RGB images, we use the VGG-16 architecture (Simonyan & Zisserman, 2014) pre-trained on ImageNet (Russakovsky et al., 2015) and fine-tuned to our data for relation detection.

**Depth Map Feature Extraction:** For depth map extraction we use ResNet18 proposed in (He et al., 2015). We trained this model from scratch following the earlier discussions in Subsection 2.1.2. We trained this network on a pure depth-based, relation detection task using Adam (Kingma & Ba, 2014), with a learning rate of $10^{-4}$ and batch size of 32 for eight epochs.

**Relation Detection Network:** Finally, given the features extracted from previous models with the location features described in Subsection 2.1.2, we trained our relation detection model. We connected each feature pair described in the previous section, separately to a fully connected hidden layer of 64, 200, 4096 and 20 neurons each, with a dropout rate of 0.1, 0.8, 0.8 and 0.1, with a scaling layer initialized as 1.0, 0.3, 0.5 and 1.0. The penultimate layer contains 4096 neurons with 0.2 dropout. We trained this network by Adam (Kingma & Ba, 2014), with a learning rate of $10^{-4}$. We used a batch size of 16 and eight epochs of training. All of the layers were initialized with Xavier weights (Glorot & Bengio, 2010). During the training of this network, RGB Feature Exraction and RGB-to-Depth network weights were frozen.

---

[3]We carried on Visual Genome experiments in an isolated framework, where all pre-trained weights and hyper-parameters, except for the depth map channel, were kept identical as the ones employed by Neural Motifs (Zellers et al., 2018). The code will be released upon publication.

Table 1: Predicate prediction recall values on VG test set. We have omitted *so* for better readability. When the depth maps are utilized together with all other features (*Ours*-l, **c**, **v**, **d**), we gain a large improvement compared to the state-of-the-art. In Macro setting, this improvement is almost as large as *Ours*-**c** to *Ours*-**c**, **v**, **l**, demonstrating the importance of depth features in relation detection. One can also see that even replacing depth maps with visual features (*Ours*-l, **c**, **d** compared to *Ours*-l, **c**, **v**) can yield better results. Additionally, comparing *Ours*-**c**, **v**, **l** to *VTransE* and *Neural Motifs* reveals the advantage of our simple model regardless of depth maps.

| | Strategy | Macro | | | Micro | | |
| | Task | Predicate Pred. | | | Predicate Pred. | | |
| | Metric | R@100 | R@50 | R@20 | R@100 | R@50 | R@20 |
| models | VTransE (Zhang et al., 2017) | - | - | - | 62.87 | 62.63 | - |
| | Yu's-S (Yu et al., 2017) | - | - | - | 49.88 | - | - |
| | Yu's-S+T (Yu et al., 2017) | - | - | - | 55.89 | - | - |
| | IMP (Xu et al., 2017) | - | - | - | 53.00 | 44.80 | - |
| | Graph R-CNN (Yang et al., 2018b) | - | - | - | 59.10 | 54.20 | - |
| | NM (Zellers et al., 2018) | 14.39 | 13.20 | 10.25 | 67.10 | 65.20 | 58.50 |
| ablations | Ours - $d$ | 9.51 | 8.46 | 6.35 | 54.72 | 51.90 | 43.86 |
| | Ours - $c$ | 15.65 | 13.09 | 8.56 | 64.82 | 60.54 | 49.89 |
| | Ours - $v$ | 13.88 | 12.24 | 8.99 | 61.72 | 58.50 | 50.41 |
| | Ours - $l$ | 5.19 | 4.66 | 3.57 | 49.07 | 46.13 | 37.48 |
| | Ours - $v, d$ | 15.47 | 14.04 | 10.83 | 62.88 | 60.52 | 53.07 |
| | Ours - $l, v, d$ | 15.76 | 14.40 | 11.07 | 63.06 | 60.83 | 53.55 |
| | Ours - $l, c, d$ | 21.67 | 19.56 | 15.12 | 67.97 | 66.09 | 59.13 |
| | Ours - $l, c, v$ | 19.16 | 17.72 | 13.93 | 67.94 | 66.06 | 59.14 |
| | Ours - $l, c, v, d$ | **22.72** | **20.74** | **16.40** | **68.00** | **66.18** | **59.44** |

## 3.3 METRICS

Some relations might not be annotated in the test set. Still, due to the model's generalization, they might get higher prediction values than the annotated ones. Therefore, similar to previous works (Nickel et al., 2016; Krishna et al., 2017), we report Recall@K (R@K). Recall per image is defined as the ratio of ground truth labels that appear in the model's top K predictions. However, as the distribution of relations in VG is highly imbalanced, this score is dominated by frequently labeled relations and might not reflect the improvements in some important but under-represented classes. Thus, we propose to compute *Macro R@K* and name the former computation metric as *Micro R@K*. For a discussion upon Micro versus Macro for F1 measure refer to (Schütze et al., 2008). In Macro R@K, we compute Micro R@K separately for each predicate class and average over it. In this setting, the prediction accuracy of under-represented classes has the chance to have a stronger effect on general accuracy.

## 3.4 COMPARING METHODS

We compare our results with *VTransE* (Zhang et al., 2017) that takes visual embeddings and projects them to relation space using TransE. We also compare to the student network of Yu et al. (2017) (*Yu's-S*), and their full model (*Yu's-S+T*) that employs external language data from Wikipedia. In the context propagating methods we report Neural Motifs (Zellers et al., 2018), Graph R-CNN (Yang et al., 2018b) and IMP Xu et al. (2017). In an ablation study, we report our relation prediction results under several settings in which different combinations of object information are employed for prediction.

## 3.5 EXPERIMENTS

Our main goal is to investigate the role of depth maps and other object information in relation detection. Therefore, we do not focus on improving the object detection accuracy and report *predicate prediction* results. In this setting, the relation detection performance is isolated from the object detector's error by using ground truth bounding boxes of the entities and their class labels to predict the most likely predicates. We carried on our experiments by training each model 8 times with different random seeds. The maximum variance was no more than 0.01. In the following we provide a discussion over the quantitative and qualitative results.

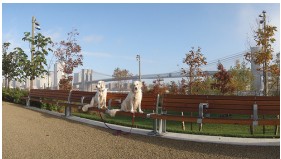 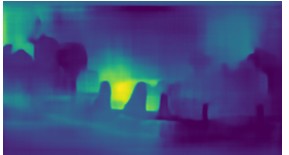 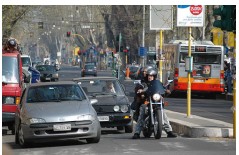 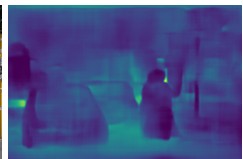

Figure 3: Using depth maps improves the understanding of *perspective* and increases the general detection rate of relations such as *behind* (e.g. right image). However, in some cases such as in the left image, after including depth maps, *(bushes, behind, bench)* has been mislabelled. One should note that here the bushes (left corner of the image) have similar depth values as the benches. Equal depth values can be a strong indicator of the predicate *next to* rather than *behind*. In this case, the human point of view seems to have caused a labelling error recovered by the depth-based model.

The complete predicate prediction results are shown in Table 1. The upper part of the table demonstrates the results directly reported from other works while the lower two parts present the results from ablation study on our model. For NM, we have computed the Macro R@K results from their available model and code (reproducing the same Micro R@K results as reported in their paper). We can see that our full model achieves the highest accuracy in comparison to the others in all settings specially under Macro R@K. It is also interesting to note that when using *only* depth maps we can already achieve a significant accuracy in predicate prediction, emphasizing the value of relational information stored within depth maps. We can observe the improvements that externally generated depth maps can bring to *Ours-v* by comparing it to *Ours-v, d*. Also comparing *Ours-c, d, l* to *Ours-c, v, l* is specially informative from two aspects: (1) It shows that while some results are almost equal in Micro settings, one can observe a significant difference under Macro settings, thus observing the changes in under-represented classes. (2) Considering that $v$ alone has a better R@K than $d$ alone when adding them separately to $c, v$ we can see that $d$ has more to offer. In other words, $v$ has more redundant information for $c, v$ compared to $d$. One should note that the improvement is not a linear function of extra information and the combination of different information sources might have non-linear effects on the outcome.

To get a better intuition of the improvements that we gain after including depth maps (*Ours-v, d*) to *Ours-v*, we plotted the changes in prediction accuracy for each predicate in Figure 4. We used darker shades for over-represented classes and lighter shades for under-represented ones. This helps to also gain a better intuition of improvement versus representativeness. We can see that in general the accuracy of relations including the predicates such as *under*, *in front of* and *behind* has been improved. However, we also observed in our data, that in some cases the predicate *behind* was mislabeled after adding the depth maps. To further explore the potential reason behind some of such successes or failures, we further examined the test images. Figure 3 gives an example of a failure and a success case in predicting *behind*. While in the right image the annotations indicate that *(bushes, behind, bench)* based on the corresponding depth map, they are in the same distance from the camera (bushes can be seen in the left corner of the image). One can argue that in fact, bushes are next to the bench rather than behind, indicating the noisy ground truth. However, it is also important to note that humans describe a predicate from their point of view which is not necessarily associated with the camera's point of view. This effect might appear in many other instances within the dataset, leading to worse performances when dealing with such predicates. A simple way to overcome such problems is by having a richer set of data.

We provide some samples of generated depth maps from VG images, with high and low quality in Figure 6 and 5. Note that the potential imprecisions within generated low-quality depth maps, e.g. sky and reflective objects such as glasses, are inevitable.

## 4 CONCLUSION

We identified 3D information as an important attribute for visual relation detection. Since this information is typically not provided in visual data sets, we employed an RGB-to-Depth network, which was pre-trained on a large corpus of data, to generate depth information. We provided a metric, *Macro R@K* for a better evaluation. In extensive empirical evaluations, we demonstrated the effect of

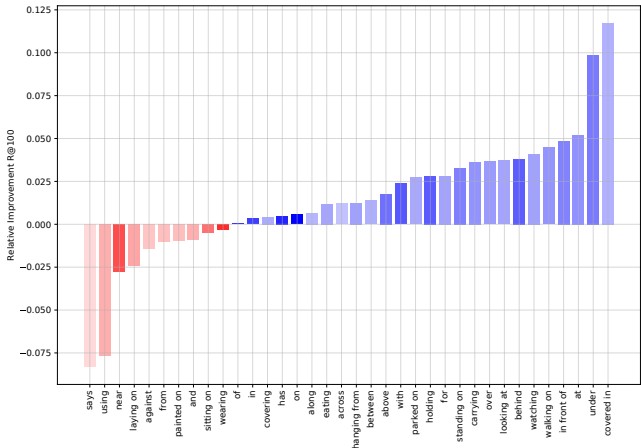

Figure 4: This plot shows the prediction changes per predicate, going from *Ours-v* to *Ours-v, d*. The classes with zero changes are omitted from the plot. The darker shades indicate larger number of that class within the test set whereas the lighter shades are under-represented classes. An improvement in predicates with more frequency has a larger effect on the Micro R@K whereas this effect is eliminated within Macro R@K. We can see that indeed the improvements by using depth maps are mostly happening within the less-represented classes.

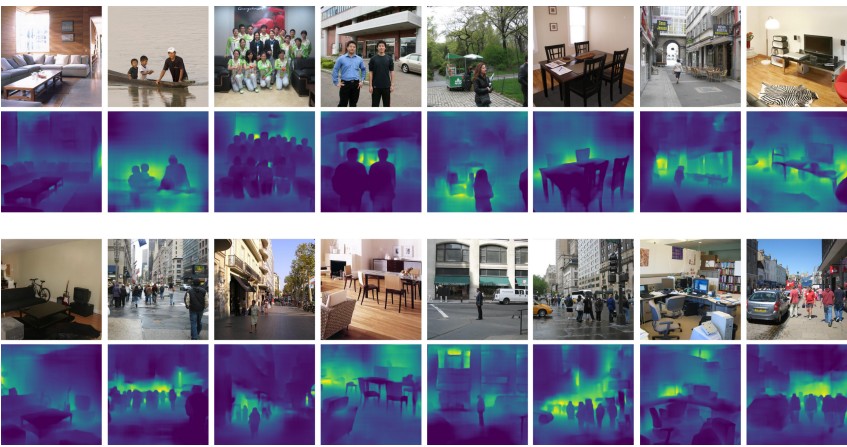

Figure 5: Examples of high quality depth maps generated from visual genome images.

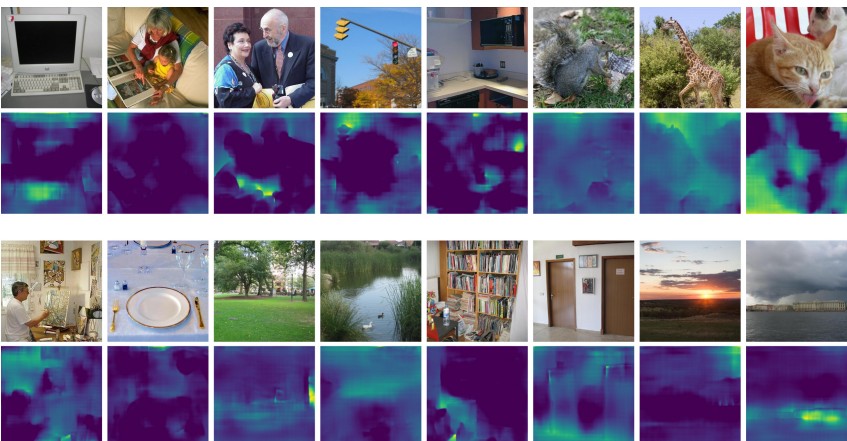

Figure 6: Examples of low quality depth maps generated from visual genome images.

different object information in visual relation detection and showed that by using depth information, one achieves significantly better performance compared to other state-of-the-art methods.

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
