# OpenReview forum: "Improving Visual Relation Detection using Depth Maps"
_ICLR.cc/2020/Conference — Reject_

### Official Review · AnonReviewer1 · 2019-10-14
**Official Blind Review #1**

**Rating:** 6

**Review:**



********* Post Rebuttal *********

I appreciate the authors' effort in providing thorough responses and revised manuscript.

I agree with the authors that "the finding not being surprising" is not a ground for rejection. I tried to word my final decision carefully but it seems it has still caused confusion for the authors. As I have mentioned in my original review, the rating was a result of the 4 points considered *together* .

That is, if one exploits privileged information that needs extra sensory data and/or annotation (point 2), *and*, this privileged information is clearly related and thus should be normally useful for the final task (point 1), *and* achieve marginal improvements (point 3), it can be a ground for rejection. Especially, given that prior works with similar arguments exists (point 4).

The rebuttal has alleviated the issue of marginal improvements (point 3) by introducing meanR@K (or as the revised paper refer to it, Macro R@K). Here, the improvements are more significant both compared to the state of the art and ablated baselines.

The authors also argue that the related [Yang et al. 2018] paper (point 4) should be considered a concurrent submission since the authors original submission was to AAAI18.

The rebuttal also addresses other clarity or experimental issues which improves the quality of the revised work.

Finally, I understand that the privileged information is only required during training time which is a good point.

All in all, *assuming that [Yang et al. 2018] is considered a concurrent work* according to ICLR, I think the revised paper becomes slightly above borderline and thus I change my rating to "weak accept". If [Yang et al. 2018] is not considered concurrent work, then, a conclusive comparison is required for the acceptance of the current work.


********* Summary *********

The paper poses the question of whether depth information is informative for visual relationship prediction using still images. It is intuitive that 3D arrangement of objects in an image can be a useful cue for predicting their relationship. As such it is important to see whether and to what extent depth information complements RGB information for visual relation detection. That is the focus of this paper.
The paper proposes to use an off-the-shelf monocular depth estimation networks to augment the available RGB information towards better visual relation detection. For that, it proposes a specific network two-stream structure working on RGB image and (predicted) depth image. The proposed model demonstrates improved results upon state of the art for visual relation prediction.


********* Strengths and Weaknesses *********

+ A comprehensive set of tests has been conducted.
+ Zero-shot prediction results are particularly interesting.
+ The experiment on ranking the predicate classes based on the change in prediction accuracy before and after using depth information (Figure 4) is interesting and intuitive.
* The final results improve upon the state-of-the-art, especially on the zero-shot learning regime. However, it seems that the improvement is mainly coming from the new architecture as opposed to the inclusion of the depth information. That is, ours_{c,v,l} brings most of the improvement already the last step to ours_{c,v,l,d} is negligible for non-zero-shot case.
- Along the same line, it’s possible that this small difference between ours_{c,v,l} and ours_{c,v,l,d} for the standard predicate prediction, can be due to a hyper-parameter optimization that is (only or more thoroughly) done for ours_{c,v,l,d}. The hyper-parameter optimization scheme for different baselines is not described.
- Given the small difference of ablation levels, the comparison will be stronger if done multiple times and reporting mean and standard deviation of the results.
- For a fair comparison the visual feature vector v_{so} should be tried as the feature of the union bound box of both subject and object same way as it is done for depth feature vector d_{so}.
- The paper refers to “Ours-d’_{so}” as a baseline that *only* uses depth information with no image/label information. However, it seems that the region proposals for this feature are coming from the image-based network that uses image information.

- Important related but uncited works:
(1) [“Visual Relationship Prediction via Label Clustering and Incorporation of Depth Information” ECCV workshops 2018] studies the same question as part of their work.


********* Final Decision *********

I do not find the paper passing the acceptance bar mainly due to the following reasons together:
1) The finding is not surprising since most of the visual relations are either explicitly depth-related (e.g., behind) or are semantically constrained by depth (e.g. riding cannot happen at different depths when the image is taken orthogonal to the rider).
2) an additional depth dataset is used which provides the model with privileged information. Should it have been the case that depth information were inferred without an additional offline dataset, the results would have been more interesting.
3) the improvements due to the additional depth network are not significant or conclusive.
4) there is a prior uncited work with the same research question for effectiveness of depth information in visual relation detection which uses a similar approach.


********* Minor points *********

- the code is not available. This is especially important since the paper is outperforming prior works which could be a contribution if reproducible.
- Section 2.2: is l_{so} concatenation of l_s and l_o?
- Section 2.2: y_{spo} is defined but never used.
- Equation 2: why do we have both e_p and f in the exponents? Aren’t they the same?
- Equation 2:  P is never defined.
- Page 5: “a fully connected hidden layer of 64, 200, 4096 and 20 neurons”: this amounts to 3 hidden layers.
- Why VGG network for visual feature and AlexNet for depth features?
- zero-shot learning results on visual genome is missing
- training procedure is a bit unclear: the text suggest that the fine tuning and/or learning of the three components might happen separately. It is important to clearly state if they are done in an end-to-end fashion and simultaneously or separately; and why.
- It’s good to name the method in table 2 in the same fashion as table 1. With the current naming (based on architecture) it is a bit confusing to understand the content without additional cross referencing. For instance AlexNet-BN - Raw seems to correspond to Ours_{c,v,l,d}
- Figure 4: the frequency represented as different shades of red or blue is really hard to notice especially on a printed paper. The red vs blue color coding is not necessary since the bars going up or down indicate the same quality. So, it might be better to use red/blue for frequency instead (e.g. dark red high frequency to dark blue low frequency)
- Section 3.2: the AlexNet reference seems wrong, it should be "ImageNet Classification with Deep Convolutional Neural Networks" NIPS , 2012
- The structure of section 3.5 is currently flat while the content seems to be nested (two experiments and two sets of corresponding discussions). It will read better if they are organized into subsections.


********* Points of extensions (improvement) *********

- I believe *unsupervised* discovery of depth information for visual relation detection can be an interesting direction since it is not limited to the availability of relevant depth dataset.
- It is not clearly motivated why one should use two separate networks for depth and RGB inputs in light of the additional complexity. For instance, it is good to discuss what is the advantage of the proposed (computationally more expensive) method over the following two simpler baselines:
- Faster RCNN is used on RGBD input to produce a single feature vector
- above case with RGB input but have the Faster RCNN predict the depth map as an auxiliary loss.




**Experience Assessment:**

I have read many papers in this area.

**Review Assessment: Checking Correctness Of Derivations And Theory:**

I carefully checked the derivations and theory.

**Review Assessment: Checking Correctness Of Experiments:**

I carefully checked the experiments.

**Review Assessment: Thoroughness In Paper Reading:**

I read the paper thoroughly.

---

> ### Author Response · Authors · 2019-11-13
> **Question regarding one of the points**
>
> Dear Reviewer 1, thank you very much for your constructive feedback. We are working on your points and will soon release the updated paper together with our responses. In the meantime, one of the points was not fully clear to us and we would like to ask you for further elaboration. This is about Point (2) in "Final Decision". If we understood correctly this point is connected to Point (1) in “Points of extensions (improvement)”. In that case, unsupervised training of RGB-to-Depth network would mean generating depth maps from RGB images without having access to any external datasets containing corresponding depth maps (as the supervised signal). The question is whether it is possible to convert one modality to another without having any parallel data (RGB and corresponding depth maps)? We do not see an obvious way how to achieve that (it would be similar to the task of learning a function that generates animal sounds by looking at their images (going from image to sound modality) without having access to any parallel image and sound data).

---

> ### Author Response · Authors · 2019-11-15
> **Response to Reviewer #1**
>
> Q1. The final results improve upon the state-of-the-art, especially on the zero-shot learning regime. However, it seems that the improvement is mainly coming from the new architecture as opposed to the inclusion of the depth information. That is, ours_{c,v,l} brings most of the improvement already the last step to ours_{c,v,l,d} is negligible for non-zero-shot case.
>
> A1. This is a correct observation. Please note that the improvement in visual relation detection community are generally in a smaller range, for example, Graph R-CNN improves the previous baseline by 1,5% points and neural motifs improve ‘no context’ baseline by 1,4% points. However, to address your concern and shed more light on this, in the updated version of the paper we provided (a) the Macro R@K measure (as mentioned) and (b) a more extensive study on the effect of each feature. Please note that the more relations are detected, the harder it gets to gain improvement with other features. The same effect happens if we assume having only c, d, l and then add v (please refer to the new ablations). In fact, depth maps can be more informative as visual features (l,c,d versus l,c,v).
>
> Q2. Along the same line, it’s possible that this small difference between ours_{c,v,l} and ours_{c,v,l,d} for the standard predicate prediction, can be due to a hyperparameter optimization that is (only or more thoroughly) done for ours_{c,v,l,d}. The hyper-parameter optimization scheme for different baselines is not described.
>
> A2. Dear Reviewer, we have reported the best possible results for each model without a focus on the full model.
>
> Q3. Given the small difference of ablation levels, the comparison will be stronger if done multiple times and reporting mean and standard deviation of the results.
>
> A3. Thank you for the suggestion. Based on your suggestion, we performed bootstrapping by training each model 8 times and updated the results. The maximum reached variance was 0.01 which we added to the text.
>
> Q4. For a fair comparison the visual feature vector v_{so} should be tried as the feature of the union bound box of both subject and object same way as it is done for depth feature vector d_{so}.
>
> A4. This is a very valid point. Using union was more of an architectural choice for us and it did not have a large effect on the results but we understand that this might have caused confusion. We re-computed the results given the concatenated subject and object depth feature vectors similar to visual features. We updated Figure 2.
>
> Q5. The paper refers to “Ours-d’_{so}” as a baseline that *only* uses depth information with no image/label information. However, it seems that the region proposals for this feature are coming from the image-based network that uses image information.
>
> A5. Region proposals for all features equally come from the ground truth as we are reporting the predicate prediction settings (since we did not want to include the image detectors error within our results). In models that do not contain l_so, we do not use region proposals as features.
>
> Q6. Important related but uncited works: (1) [“Visual Relationship Prediction via Label Clustering and Incorporation of Depth Information” ECCV workshops 2018] studies the same question as part of their work.
>
> A6. This point has been explained later in the feedback to “Final Decision”.

---

> ### Author Response · Authors · 2019-11-15
> **Response Page 2**
>
>
> ********* Final Decision *********
>
> Q1. I do not find the paper passing the acceptance bar mainly due to the following reasons together:
> 1) The finding is not surprising since most of the visual relations are either explicitly depth-related (e.g., behind) or are semantically constrained by depth (e.g. riding cannot happen at different depths when the image is taken orthogonal to the rider).
>
> A1. We respectfully disagree that the lack of a ‘surprising’ effect in a finding should be the reason for its rejection. Let us consider Neural Motifs and Graph R-CNN: getting improvements by contextualizing the object embeddings is actually a trivial idea, or in case of Yu’s work: getting improvements by employing the prior knowledge signal from a teacher network also might seem like a good idea. What matters in all the above cases is observing that such properties are missing in the current models, proposing some way to employ them while tackling the possible challenges on the way, and providing a study to extensively study their implications.
>
> Q2. an additional depth dataset is used which provides the model with privileged information. Should it have been the case that depth information was inferred without an additional offline dataset, the results would have been more interesting.
>
> A2. Dear Reviewer 1, thank you very much for your constructive feedback. As mentioned in an earlier comment, this point was not fully clear to us and we would like to ask you for further elaboration. If we understood correctly this point is connected to Point (1) in “Points of extensions (improvement)”. In that case, unsupervised training of the RGB-to-Depth network would mean generating depth maps from RGB images without having access to any external datasets containing corresponding depth maps (as the supervised signal). The question is whether it is possible to convert one modality to another without having any parallel data (RGB and corresponding depth maps)? We do not see an obvious way how to achieve that (it would be similar to the task of learning a function that generates animal sounds by looking at their images (going from image to sound modality) without having access to any parallel image and sound data).
>
>
>
> Q3. the improvements due to the additional depth network are not significant or conclusive.
>
> A3. This is an important point. We included a more extensive ablation study within the updated version of our paper as well as a more intuitive evaluation metric (Macro R@k) so that it would be easier to spot the improvements when using depth maps.
>
> Q4. there is a prior uncited work with the same research question for the effectiveness of depth information in visual relation detection which uses a similar approach.
>
> A4. Thank you for pointing us to a very relevant work. We included it in our updated version and compared it to our work. We consider this a parallel work as it has been published after the initial submission of our work to AAAI 2018. Some visible differences are:
> (1) Their feature extraction strategy is very limited (average over mask).
> (2) This work only studies human-centric relations whereas we study on a much larger and extensive dataset with a broader range of possible relations.
> (3) The experiments of this study are very limited whereas we provide an extensive set of ablation studies and comparisons to relevant works in this area.
> Nevertheless, we cited their works and added the differences and changed the wording of our paper to better reflect our contributions.

---

> ### Author Response · Authors · 2019-11-15
> **Response Page 3**
>
>
> ********* Minor points *********
>
> - the code is not available. This is especially important since the paper is outperforming prior works which could be a contribution if reproducible.
>
> Dear reviewer, the code will be made available as a fork from Neural Motifs code, upon the acceptance of the paper.
>
> - Section 2.2: is l_{so} concatenation of l_s and l_o?
> Thank you for mentioning this. Yes, l_{so} is the concatenation of l_s and l_o, and we have updated the text to clarify this point
>
> - Section 2.2: y_{spo} is defined but never used.
> Since we assumed the definition of the error function would be trivial, we didn’t explicitly use y_{spo} and correspondingly y_hat{spo} in a separate formula. To fix this issue we have removed this variable in the updated version of the paper.
>
> - Equation 2: why do we have both e_p and f in the exponents? Aren’t they the same?
>
> Fixed.
>
> - Equation 2: P is never defined.
> There was a missing \mathbf. P is defined at the beginning of that section as the set of all predicates.
>
> - Page 5: “a fully connected hidden layer of 64, 200, 4096 and 20 neurons”: this amounts to 3 hidden layers.
> Please note that each of the mentioned layers is only connected to their corresponding feature pairs separately and they are not sequentially connected to each other. We made it more clear within the text.
>
> - Why VGG network for visual feature and AlexNet for depth features?
>
> We can use pre-trained weights of VGG for better performance and we also rely on this from Neural Motifs provided weights so we are easily comparable. For depth maps since we train everything from scratch, AlexNet (or in this version ResNet18), are much easier to train and require fewer data.
>
> - zero-shot learning results on the visual genome is missing.
> Please note that the visual genome dataset doesn’t provide the zero-shot evaluations.
>
> - training procedure is a bit unclear: the text suggest that the fine tuning and/or learning of the three components might happen separately. It is important to clearly state if they are done in an end-to-end fashion and simultaneously or separately; and why.
> Thanks for the feedback. We updated the text. RGB-To-Depth network and RGB feature extraction network weights are frozen for stability. This is a common practice to keep the weights from earlier layers frozen. Weights from the other networks are not frozen. (what about depth?)
>
> - It’s good to name the method in table 2 in the same fashion as table 1. With the current naming (based on architecture) it is a bit confusing to understand the content without additional cross referencing. For instance AlexNet-BN - Raw seems to correspond to Ours_{c,v,l,d}
>
> Not applicable in the new version.
>
> - Figure 4: the frequency represented as different shades of red or blue is really hard to notice especially on a printed paper. The red vs blue color coding is not necessary since the bars going up or down indicate the same quality. So, it might be better to use red/blue for frequency instead (e.g. dark red high frequency to dark blue low frequency)
>
> Fixed.
>
> - Section 3.2: the AlexNet reference seems wrong, it should be "ImageNet Classification with Deep Convolutional Neural Networks" NIPS , 2012
> Not applicable in the new version. Anyhow please note that we used an enhanced version of AlexNet with fewer parameters and minor changes which was proposed in that paper. However, in our current evaluations, we replace AlexNet with ResNet18 which is more efficient.
>
> - The structure of section 3.5 is currently flat while the content seems to be nested (two experiments and two sets of corresponding discussions). It will read better if they are organized into subsections.
>
> Not applicable in the new version.

---

> ### Author Response · Authors · 2019-11-15
> **Response Page 4**
>
>
> ********* Points of extensions (improvement) *********
>
> Q1. I believe *unsupervised* discovery of depth information for visual relation detection can be an interesting direction since it is not limited to the availability of relevant depth dataset.
>
> A1. As mentioned in our first comment, unsupervised training of RGB-to-Depth network would mean generating depth maps from RGB images without having access to any external datasets containing corresponding depth maps (as the supervised signal). The question is whether it is possible to convert one modality to another without having any parallel data (RGB and corresponding depth maps)? We do not see an obvious way how to achieve that (it would be similar to the task of learning a function that generates animal sounds by looking at their images (going from image to sound modality) without having access to any parallel image and sound data).
>
> Q2. It is not clearly motivated why one should use two separate networks for depth and RGB inputs in light of the additional complexity. For instance, it is good to discuss what is the advantage of the proposed (computationally more expensive) method over the following two simpler baselines:
> - Faster RCNN is used on RGBD input to produce a single feature vector
>
> A2.Our focus was not on improving object detection using depth maps which is already a well-explored area. Bringing depth to the Faster R-CNN input would mostly affect object detection and we wanted to isolate this effect from visual relation detection. Also, in this case, we either need to (1) apply shared weights for RGB and D signals which is not a good idea as discussed in Section 2.1.2 or (2) use separate weights for RGB and Depth maps. This would be similar to our current architecture.
> Q3. Above case with RGB input but have the Faster RCNN predict the depth map as an auxiliary loss.
> A3. We cannot use depth maps in the loss function as we do not have access to any ground truth depth maps.

---

### Official Review · AnonReviewer2 · 2019-10-24
**Official Blind Review #2**

**Rating:** 3

**Review:**

This paper proposes to leverage the depth information for relation prediction, arguing that the depth information benefit the prediction of some predicates. To solve the lack of 3D data, an RGB-to-Depth model is trained on external available dataset and then applied to images from visual relation dataset. In the experiments, they investigate different strategies to extract features from depth maps and the explore effectiveness of depth information by comparing the model that only used depth map as input with those which use RGB information. The comparisons with other methods and ablation studies under both Zero-shot setting and normal setting demonstrate the effectiveness of depth information.

+Strength:
(1) The motivation is reasonable and what the authors make an attempt to explore is very meaningful. Visual relation especially the spatial relation is not likely to be predicted accurately without 3D information. In other words, it seems that visual relation prediction task will be extended to 3D images rather than staying within 2D images. Thus what the authors do is a good exploration for further extensions.
(2) Comparisons with previous methods and the results show that the depth information is useful to some extent, but not so obvious.
(3) The writing of this article is good and it’s very easy to understand.

-Weakness:
(1) The RGB-to-Depth Network is pretrained on other dataset. Is there any gap when it is used for VG or VRD dataset?
(2) Although the depth map feature extraction seems to work well, it seems to be a little trivial. Why a CNN, e.g. AlexNet, or VGG, can be used to extract depth features? And why the AlexNet trained from scratch performs better than AlexNet pretrained on RGB images for object detection task and VGG net? If the author can give more explanations, this part will be more insightful.
(3) From the plot which shows the top 10 percent absolute changes in prediction performance per predicate, the advantage of Depth is not obvious compared with RGB. And Depth does not bring the advantage claimed in Abstract. It’s a little hard to understand why depth information can rectify the prediction of (Tower, taller, trees). To sum up, the qualitative results are not so satisfying.
(4) In Table 1, what really functions seems to be c_so, v_so, and l_so, while the improvement brought by depth is limited.

**Experience Assessment:**

I have published one or two papers in this area.

**Review Assessment: Checking Correctness Of Derivations And Theory:**

I carefully checked the derivations and theory.

**Review Assessment: Checking Correctness Of Experiments:**

I carefully checked the experiments.

**Review Assessment: Thoroughness In Paper Reading:**

I read the paper at least twice and used my best judgement in assessing the paper.

---

> ### Author Response · Authors · 2019-11-15
> **Response to Reviewer #2**
>
> Q1. The RGB-to-Depth Network is pre-trained on other dataset. Is there any gap when it is used for VG or VRD dataset?
>
> A1. Thank you very much for your constructive feedback. Please note that we are interested in the role that depth maps can play in relation detection and evaluating the generalizability power of an RGB-to-Depth model from one dataset to another, is not the focus of our research work. Nevertheless, we updated our paper to provide samples of the generated depth maps from our datasets, as a qualitative measure. Please note that providing quantitative measures in our case is not possible as our dataset contains only RGB images and there are no ground truth depth maps available. However, you can find the quantitative measures on the test data from NYU dataset available in [1] that are supposed to be generalizable to unseen examples.
>
> [1] Laina, Iro, et al. "Deeper depth prediction with fully convolutional residual networks." 2016 Fourth international conference on 3D vision (3DV). IEEE, 2016.
>
> Q2.1. Although the depth map feature extraction seems to work well, it seems to be a little trivial. Why a CNN, e.g. AlexNet, or VGG, can be used to extract depth features?
>
> A2.1. Thanks for the interesting question. Please note that we only use the architectural design of AlexNet (VGG or ResNet18) and train the weights of the network from scratch. Any other CNN architecture would make a good feature extractor for depth maps as (similar to RGB images), in depth maps: 1) there is high covariance within the local neighborhood which diminishes with distance and 2) the statistics are mostly stationary across the depth maps. CNNs are designed with the imposed inductive bias of locality and translation invariance which perfectly exploits such characteristics of the input domain[1].
>
> [1] Battaglia, Peter W., et al. "Relational inductive biases, deep learning, and graph networks." arXiv preprint arXiv:1806.01261 (2018).
>
> Q2.2. And why the AlexNet trained from scratch performs better than AlexNet pre-trained on RGB images for object detection task and VGG net? If the author can give more explanations, this part will be more insightful.
>
> A2.2 This has been discussed briefly in Section 2.1.2. While RGB and depth maps share some characteristics (mentioned above) that make them good candidates for CNN-based feature extraction, they are still different modalities representing different information and even having a different pixel range. Therefore, sharing the same CNN weights between them would be sup-optimal. We can elaborate more on this within the text.
>
> Q3. From the plot which shows the top 10 percent absolute changes in prediction performance per predicate, the advantage of Depth is not obvious compared with RGB. And Depth does not bring the advantage claimed in Abstract. It’s a little hard to understand why depth information can rectify the prediction of (Tower, taller, trees). To sum up, the qualitative results are not so satisfying.
>
> A3. Thanks for pushing us towards more clarity. You are right. One of the points we wanted to make here was that improvements in under-represented predicates do not get reflected within the overall R@K. To address your concern regarding this, instead of only providing qualitative reports, we now report the result using a better quantitative metric (Macro R@K) which computes the R@K for each predicate separately and reports the mean overall. Please find these results in the updated version. We also further updated the mentioned plot and tried to make it more clear.
> Regarding the predicate “taller”, we removed this example as we had to remove VRD results for space constraints. However, the explanation goes like this: a shorter person standing closer to the camera can look taller than a tall person standing further away (perspective). Having access to a depth map helps us tackle this problem.
>
> Q4. In Table 1, what really functions seems to be c_so, v_so, and l_so, while the improvement brought by depth is limited.
>
> A4. This is a correct observation. Please note that the improvement in visual relation detection community are generally in a smaller range, for example, Graph R-CNN improves the previous baseline by 1,5% points and neural motifs improve ‘no context’ baseline by 1,4% points. However, to address your concern and shed more light on this, in the updated version of the paper we provided (a) the Macro R@K measure (as mentioned) and (b) a more extensive study on the effect of each feature. Please note that the more relations are detected, the harder it gets to gain improvement with other features. The same effect happens if we assume having only c, d, l and then add v (please refer to the new ablations). In fact, depth maps can be more informative as visual features (l,c,d versus l,c,v).

---

### Official Review · AnonReviewer3 · 2019-10-28
**Official Blind Review #3**

**Rating:** 3

**Review:**

OVERVIEW:
The authors propose to use depth information to better predict the visual relation between objects in an image. They do this by incorporating a pre-trained RGB-to-Depth model within existing frameworks. They claim the following contributions:
1. First to utilize 3D information in visual relation detection. They synthesize depth images for existing benchmark datasets of VRD and VG using a pre-trained RGB-to-Depth model trained on NYUv2 to generate RGB-D data for visual relation detection.
2. Discuss and empirically investigate different strategies to extract features from depth maps for relation detection.
3. Study the quantitative and qualitative benefits of incorporating depth maps. "We show in our empirical evaluation using the VRD and VG datasets, that models using depth maps can outperform competing methods by a margin of up to 3% points".

MAJOR COMMENTS:
1. I liked the idea of using depth information to inform visual relationships but I am not sure if the proposed approach is the way to go. Given a depth image of the scene, we can generate a reconstruction of the scene in 3D, even if it is partial/imperfect. Direct reasoning in 3D should now be possible instead of going via deep networks as proposed in the paper. I believe a direct 3D approach would make a meaningful baseline at the very least and needs to be discussed.
2. The authors use a pre-trained RGB-to-Depth network trained on NYU-v2 to predict depth for the images of VRD and VG. There is very little discussion about the quality of predicted depth maps. Ideally, this needs to be quantified to convince the reader that the generated depth maps are "good" but at the very least the authors need to show qualitative examples (both good, typical and bad) to prove that the pre-trained network generates meaningful depth maps.
3. To use a siamese (shared weights) feature extractor between RGB and Depth images or not, is not a significant contribution by itself. In principle, separate feature extractors lead to larger model complexity/learning capability and make sense given domain separation between RGB and Depth.

MINOR COMMENTS:
1. Figure 2 seems to indicate that a Faster-RCNN is used on both RGB and Depth steams which is backed up by text in Section 2 (first paragraph). However, in Section 3.2, under RGB Feature Extraction and Depth Map Feature Extraction, the discussion is about VGG-16 and AlexNet-BN networks. The VGG-16 network is pre-trained in ImageNet and finetuned to relevant data but it is not clear for what task? If the task is object detection, it needs to be trained for it (not fine-tuned, unless it is being initialized from COCO pre-training). The AlexNet-BN depth model is trained for relation detection using only depth. But it is not clear if it is using proposals/boxes generated by RGB detection model or using ground-truth boxes. Basically, the object-detection component of the pipeline is not clear at all.

NOTE:
I would like to mention that I have published in monocular object pose estimation and work in the object recognition. I am not as familiar with the visual relation detection field but I understand all the components proposed by the authors in this work. I believe I understood the paper and reviewed it fairly (to the best of my ability).

**Experience Assessment:**

I do not know much about this area.

**Review Assessment: Checking Correctness Of Derivations And Theory:**

N/A

**Review Assessment: Checking Correctness Of Experiments:**

I assessed the sensibility of the experiments.

**Review Assessment: Thoroughness In Paper Reading:**

I read the paper thoroughly.

---

> ### Author Response · Authors · 2019-11-15
> **Response to Reviewer #3**
>
> Q1. I am not sure if the proposed approach is the way to go. Given a depth image of the scene, we can generate a reconstruction of the scene in 3D...
>
> A1. Thank you for your valuable comments. You are right that direct reasoning in 3D would be more beneficial. However, to reconstruct a scene in 3D we would need to collect either 1) more than one RGB image or 2) more than one depth map, coming from different views of that scene. This is a more involved process that has limitations both on data and computation. What we have in our scenario is only one RGB image from the scene which we use to synthetically generate the depth map from. Going through the RGB-to-Depth deep network in our architecture is the only way to acquire those depth maps in first place. Other than the limitation in our current datasets, please note that in many real-world scenarios for humans (or for autonomous agents such as self-driving cars) this is a similar case: a car driving directly forward towards a pedestrian has no access to the rear-view of that person. Depth maps, in this case, can already provide sufficient data on the distance to objects. In summary, we are exploring a real-world scenario where a 3D reconstruction of the scene is not accessible.
>
> 2. There is very little discussion about the quality of predicted depth maps. Ideally, this needs to be quantified...
>
> A2. Thank you for the nice suggestion. Please find the attached qualitative examples in the updated version of our paper. Please note that providing quantitative measures in our case is not possible as our dataset contains only RGB images and there are no ground truth depth maps available. However, you can find the quantitative measures on the test data from the NYU dataset available in [1] that are supposed to be generalizable to unseen images.
>
> [1] Laina, Iro, et al. "Deeper depth prediction with fully convolutional residual networks." 2016 Fourth international conference on 3D vision (3DV). IEEE, 2016.
>
> 3. To use a siamese (shared weights) feature extractor between RGB and Depth images or not, is not a significant contribution by itself...
>
> A3. Thank you for pointing this out. We considered this a contribution as most state-of-the-art works assume otherwise without providing sufficient experiments on it. However, we updated our contributions list to reflect your concern. We removed this item as our contributions and only described it briefly in the feature extraction section.
>
> Minor Comments:
>
> 1. Figure 2 seems to indicate that a Faster-RCNN is used on both RGB and Depth steams which is backed up by text in Section 2 (first paragraph)...
>
> A1. We did not aim to indicate that. Please note that Faster-RCNN has the region proposal networks and the network we apply to depth stream in Figure 2 has only a feature extractor (ResNet18 in this case). As shown in the RGB stream, RPN (from Faster R-CNN) is only applied to RGB images and the extracted regions are also used to provide bounding boxes for the depth maps (also explained in the mentioned paragraph).
>
>
> 2. The VGG-16 network is pre-trained in ImageNet and finetuned to relevant data but it is not clear for what task? If the task is object detection, it needs to be trained for it (not fine-tuned, unless it is being initialized from COCO pre-training)...
>
> A2. This is pre-trained for object detection and fine-tuned for predicate prediction. We appreciate your comment. We made it clearer in the paper.
>
> 3. The AlexNet-BN depth model is trained for relation detection using only depth. But it is not clear if it is using proposals/boxes generated by RGB detection model or using ground-truth boxes. Basically, the object-detection component of the pipeline is not clear at all.
>
> A3.We use the ground-truth bounding boxes. During training, using noisy proposals from RGB detection would be sub-optimal (specially under predicate prediction setting).

---

### Author Response · Authors · 2019-11-15
**To all reviewers**

We thank the reviewers for their constructive feedback. We have revised the paper by taking into account most of the mentioned concerns.
(Additions)
The main mutual concern was regarding the improvement percentage (Reviewer 1 and 2). While we would like to mention that the improvements in visual relation detection community are generally in a smaller range (for example Graph R-CNN improves the previous baseline by 1,5% points and Neural Motifs improves ‘no context’ baseline by 1,4% points), we addressed this concern by updating our paper as follows: (1) we provided a more extensive ablation study. (2) we extended our qualitative results about the under-represented predicates (Figure 4), to quantitative results by proposing to use a more competent metric (Macro R@K). This metric takes into account the improvements within each predicate class individually and can give a better intuition about the results.
We also added the qualitative results on the generated depth maps to the paper, as requested by Reviewer 3 and 2.

(Removals)
We removed the 3rd contribution regarding the feature extraction strategy as suggested by reviewer 3.
We removed the results of the VRD dataset as new metric (Macro R@K) evaluations limited our space. VRD dataset is less often employed in state-of-the-art works and we believe this would not hurt our contribution. We updated all the figures accordingly. Please note that Figure 1 is an image both in VRD and VG datasets.

---

### Author Response · Authors · 2020-02-02
**New Version After ICLR Decision**

We thank all the ICLR reviewers and meta-reviewer again for their constructive feedback. We encourage the interested readers to follow our discussions during the rebuttals.

We made a large revision during the rebuttal period, including a new contribution, however, the changes seem to have been only noted by reviewer 1, and unfortunately, there was no additional feedback from other reviewers. Nevertheless, by considering all the comments one more time, we further clarified some points and made new changes to our paper. We released the VG-Depth dataset, our code and an updated version of the paper:

Code and Dataset: https://github.com/Sina-Baharlou/Depth-VRD
Updated Paper: https://arxiv.org/abs/1905.00966


---------------------------------------------------------------------------------------------------------------------

Specifically, considering the points by meta-reviewer, we made the following changes:

1. We clarified this in our new version. We do not see any advantages in using a unified architecture when different modalities are involved. For example, if one is dealing with sound data in parallel to RGB images, it is reasonable to have an RNN and a CNN feature extractor. The same applies to depth maps and RGB images. Employing VGG to extract RGB features gives us the benefit of transfer learning (from pre-trained models). On the other hand, since we cannot do the same for depth maps, the choice of ResNet gives us the advantage of training an efficient model with less training data.
In our new version, we focused furthermore on studying (1) the importance of different features for relation detection, and (2) whether the evaluation metrics can properly reflect that. Please note the various quantitative and qualitative results reported in the post-rebuttals paper.

2. We provided more qualitative results in the new version of our paper. Providing quantitative measures in our case is not possible as our dataset contains only RGB images and there are no ground truth depth maps available. This is the reason to release the synthetic VG-Depth dataset. One can find the quantitative measures on the test data of NYU dataset available in [1], generalizable to unseen images such as the ones in Visual Genome. The work from 2016 is still one of the only open-source methods that have competitive results for RGB-to-Depth generation and as shown in our study, this is already giving a large boost to our detection rate. The main focus is neither on evaluating nor on engineering a better RGB-to-Depth model, but rather on studying its applicability in improving visual relation detection.

3. We took AC’s comment into account, and provided an additional paragraph in our new version discussing this prior work.

4. As mentioned in the rebuttals discussion with reviewers 3, we agree that direct reasoning in 3D would be more beneficial. However, to reconstruct a scene in 3D we would need to collect either (1) more than one RGB image or (2) more than one depth map, coming from different views of that scene. This is a more involved process that has limitations both on data and computation. What we have in our scenario is only one RGB image from the scene which we use to synthetically generate the depth map from. Going through the RGB-to-Depth deep network in our architecture is the only way to acquire those depth maps in the first place. Other than the limitation in our current datasets, please note that in many real-world scenarios for humans (or for autonomous agents such as self-driving cars) this is a similar case: a car driving directly forward towards a pedestrian has no access to the rear-view of that person. Depth maps, in this case, can already provide sufficient data on the distance to objects. In summary, we are exploring a real-world scenario where a 3D reconstruction of the scene is not accessible.

[1] Laina, Iro, et al. "Deeper depth prediction with fully convolutional residual networks." 2016 Fourth international conference on 3D vision (3DV). IEEE, 2016.

---

### Decision · Program_Chairs · 2019-12-19

**Decision:**

Reject

**Comment:**

The paper proposes to improve visual relation prediction by using depth maps.  Since existing RGB images do not contain depth informations, the authors use a monocular depth estimation method to predict depth maps.  The authors show that using depths maps, they are able to improve prediction of relations between ground truth object bounding boxes and labels.

The paper got relatively low scores (with 3 initial weak rejects).  After the revision and suggested improvements, one of the reviewers updated their score so the paper now has 2 weak rejects and 1 weak accept.

The paper had the following weaknesses:
1. The paper has limited technical novelty as it combines off the shelf components.  The components also used different backbones (ResNet at some places, VGGNet at others) that were directly from prior work.  Was there any attempt to have an unified architecture? As the main novelty of the work is not in the model aspect, the paper needs to have stronger experiments and analysis.
2. More analysis on the quality of the depth estimation is needed.  Ideally, the work should provide some insight into whether some of the errors is due to having bad depth estimation?  The depth estimation method used is from 2016, there are newer depth estimation methods now.  Would having better depth estimation give improved results?  Experiments that illustrates that method works well with predicted bounding boxes instead of ground truth bounding boxes will also strengthen the paper.
3. There was the question of whether the related Yang et al. 2018 workshop paper should be included as basis for comparison.  In the AC's opinion, Yang et al. 2018 is not concurrent work and should be treated as prior work.  However, it is not clear whether it is feasible to compare against that work.  The authors should attempt to do so and if infeasible, clearly articulate why that is the case.
4. As pointed out by R3, once there is a depth map available, it is also possible to compare against 3D methods (such as those that operate on point clouds)

Overall the paper had a nice insight by proposing the simple but effective idea of using depth information to help with visual relation prediction.  Still the work is somewhat borderline in quality.  In the AC's opinion, the main contribution and insight of the paper is of limited interest to the ICLR community, and it would be more appreciated in a computer vision conference.  The authors are encouraged to improve the paper with stronger experiments and analysis, incorporate various suggestions from the reviewers, and resubmit to a vision conference.